# Osteoblast-osteoclast co-cultures: A systematic review and map of available literature

Stefan J. A. Remmers[1], Bregje W. M. de Wildt[1], Michelle A. M. Vis[1], Eva S. R. Spaander[1], Rob B. M. de Vries[2], Keita Ito[1], Sandra Hofmann[1]*

1 Department of Biomedical Engineering and the Institute of Complex Molecular Systems, Orthopaedic Biomechanics, Eindhoven University of Technology, Eindhoven, The Netherlands, 2 Department for Health Evidence, SYRCLE, Radboud Institute for Health Sciences, Radboudumc, Nijmegen, The Netherlands

☯ These authors contributed equally to this work.
* s.hofmann@tue.nl

**Data Availability Statement:** All relevant data are within the manuscript and its Supporting information files.

## Abstract

Drug research with animal models is expensive, time-consuming and translation to clinical trials is often poor, resulting in a desire to replace, reduce, and refine the use of animal models. One approach to replace and reduce the use of animal models is to use *in vitro* cell-culture models. To study bone physiology, bone diseases and drugs, many studies have been published using osteoblast-osteoclast co-cultures. The use of osteoblast-osteoclast co-cultures is usually not clearly mentioned in the title and abstract, making it difficult to identify these studies without a systematic search and thorough review. As a result, researchers are all developing their own methods, leading to conceptually similar studies with many methodological differences and, as a consequence, incomparable results. The aim of this study was to systematically review existing osteoblast-osteoclast co-culture studies published up to 6 January 2020, and to give an overview of their methods, predetermined outcome measures (formation and resorption, and ALP and TRAP quantification as surrogate markers for formation and resorption, respectively), and other useful parameters for analysis. Information regarding these outcome measures was extracted and collected in a database, and each study was further evaluated on whether both the osteoblasts and osteoclasts were analyzed using relevant outcome measures. From these studies, additional details on methods, cells and culture conditions were extracted into a second database to allow searching on more characteristics. The two databases presented in this publication provide an unprecedented amount of information on cells, culture conditions and analytical techniques for using and studying osteoblast-osteoclast co-cultures. They allow researchers to identify publications relevant to their specific needs and allow easy validation and comparison with existing literature. Finally, we provide the information and tools necessary for others to use, manipulate and expand the databases for their needs.

**Funding:** S.J.A.R. was supported by ZonMw More Knowledge with Fewer Animals Programme (MKMD), project number 114024141 (https://www.zonmw.nl). S.J.A.R. and S.H. were financially supported by the European Union's Seventh Framework Programme (FP/2007-2013), Grant Agreement No. 336043 (project REMOTE) (https://erc.europa.eu). B.W.M.d.W., M.A.M.V. and S.H. are financially supported by the research program TTW with project number TTW 016.Vidi.188.021, which is (partly) financed by the Netherlands Organization for Scientific Research (NWO) (https://www.nwo.nl/). The funders had no role in study design, data collection and analysis, decision to publish, or preparation of the manuscript.

**Competing interests:** The authors have declared that no competing interests exist.

## Introduction

Bone is a highly dynamic tissue with mechanical and metabolic functions that are maintained by the process of bone remodeling by bone forming osteoblasts (OBs), bone resorbing osteoclasts (OCs), and regulating osteocytes. In healthy tissue, bone resorption and formation are in equilibrium, maintaining the necessary bone strength and structure to meet the needs of the body. In diseases such as osteoporosis and osteopetrosis this equilibrium is disturbed, leading to pathological changes in bone mass that adversely affect the bone's mechanical functionality [1].

Studies on bone physiology, bone disease and drug development are routinely performed in animal models, which are considered a fundamental part of preclinical research. The use of animals raises ethical concerns and is generally more time consuming and expensive than *in vitro* research. Laboratory animals are also physiologically different from humans. Their use in pre-clinical studies often leads to poor translation of results to human clinical trials [2, 3] and subsequent failure of promising discoveries to enter routine clinical use [4, 5]. These limitations and the desire to reduce, refine and replace animal experiments gave rise to the development of *in vitro* models [6, 7]. Over the last four decades, significant progress has been made towards developing OB-OC co-culture models.

The development of *in vitro* OB-OC co-cultures started with a publication of T.J. Chambers in 1982 [8], where the author induced quiescence of isolated tartrate resistant acid phosphatase (TRAP)-positive rat OCs with calcitonin and reversed their quiescence by co-culturing them with isolated rat OBs in direct contact. At that time, studies involving OCs resorted to the isolation of mature OCs by disaggregation from fragmented animal bones. The first account of *in vitro* osteoclastogenesis in co-culture was realized in 1988 when Takahashi and co-authors [9] cultured mouse spleen cells and isolated mouse OBs in the presence of 1α,25-dihydroxyvitamin D3 and found TRAP-positive dentine-resorbing cells. The herein described methods were used and adapted to generate OCs for the following decade. Most of the studies published until this point in time used co-cultures as a tool for achieving osteoclastogenesis, as opposed to a model for bone remodeling. At that time, a co-culture of OBs with spleen cells or monocytes was the only way of generating functional OCs *in vitro*. It wasn't until 1999 that Suda [10] discovered Receptor Activator of Nuclear Factor Kappa Ligand (RANKL) and Macrophage Colony Stimulating Factor (M-CSF) as the necessary and sufficient proteins required for differentiating cells from the monocyte/macrophage lineage into functioning OCs [11–13]. This discovery marked the start of co-culture models developed for studying bone remodeling.

In recent years, many research groups have ventured into the realm of OB-OC co-cultures with the intent of studying both formation and resorption, but each group seems to be individually developing the tools to suit their needs resulting in many functionally related experiments that are methodologically completely different. In addition, the use of such methods is often not clearly stated within title and abstracts. Simple title/abstract searches such as 'OB + OC + co-culture' show only a fraction of available studies using OB-OC co-cultures. Finding and comparing different co-culture approaches and their results is thus complicated which forces each group to develop and use their own methods.

The aim of this study was to conduct a systematic review of all OB-OC co-cultures published up to January 6, 2020. With this systematic review, we aimed at identifying all existing OB-OC co-culture studies and analyze these within two comprehensive databases, allowing researchers to quickly search, sort and select studies relevant for their own research. Database 1 contains all OB-OC co-culture studies in which at least one relevant primary outcome measure was investigated (formation and/or resorption) or secondary outcome measure (alkaline phosphatase (ALP) and/or tartrate resistant acid phosphatase (TRAP) quantification as

surrogate markers for formation and resorption, respectively) (S1 File). A sub-selection of studies that investigated these relevant outcome measures on both OBs and OCs in the co-culture was included in Database 2, accompanied by additional details on methods, culture conditions and cells (S2 File). The collection of the two databases will further be referred to as a systematic map.

## Methods

For this systematic map a structured search protocol was developed using the SYRCLE protocol format [14]. The protocol and search strings were made publicly available before completion of study selection via Zenodo [15] to ensure transparency of the publication. In short, three online bibliographic literature sources were consulted with a comprehensive search query and the resulting publications were combined and screened using a four-step procedure (Fig 1): 1) identification of OB-OC co-cultures, 2) identification of relevant outcome measures, 3) categorization in Databases 1 and 2 (Fig 2), 4) search for additional articles in the reference lists of studies included in Database 2 and relevant reviews.

### Database search

The online bibliographic literature sources Pubmed, Embase (via OvidSP) and Web of Science were searched on January 6, 2020 with a predefined search query consisting of the following components: ([OBs] OR ([OB precursors] AND [bone-related terms])) AND ([OCs] OR ([OC precursors] AND [bone-related terms])) AND [co-culture], where each component in square brackets represents a list of related thesaurus and free-text search terms. The full search strings can be found via Zenodo [15]. The results of all three searches were combined. Conference abstracts and duplicates were removed using the duplicate removal tools of Endnote X7 and Rayyan web-based systematic review software [16]. The entire screening and data collection process was performed independently by two researchers.

**Screening step 1: Identification of OB-OC co-cultures.** This step was performed to identify and extract OB-OC co-cultures from the complete list of studies identified from the three online bibliographic literature sources after automatic removal of conference abstracts and duplicates. Using Rayyan web-based systematic review software [16], the titles and abstracts were screened for the presence of primary studies using OB-OC co-cultures. Reviews, theses, chapters, and conference abstracts that were not automatically detected were excluded at this point. Potentially relevant reviews were saved separately to serve as an additional source of studies that could have been missed by the systematic search.

In the selection process, co-culture was defined as the simultaneous (assumed) presence of OBs and OCs (or OB-like and/or OC-like cells) within the same culture system at a moment during the described experiment such that the cells were able to communicate either via soluble factors in the medium and/or direct cell-cell contact. Both primary cells and cell lines of any origin were admitted including heterogeneous cell populations if these were clearly defined and expected to result in a biologically relevant number of the desired cell type. The presence of progenitor cells (such as monocytes or mesenchymal stem/stromal cells) was allowed only if these were either verified or expected to differentiate into OBs and/or OCs. Studies using a single animal or human donor for both cell types were allowed, but only if the two (progenitor) cell types were at one point separated, counted, and reintroduced in a controlled manner. Trans-well systems (no physical contact but shared medium compartment with or without membrane), scaffolds (3-dimensional porous structure of any material including decellularized matrix), and bioreactor culture systems (culture exposed to physical stimuli such as rotation, mechanical loading or fluid flow) were included. Conditioned media

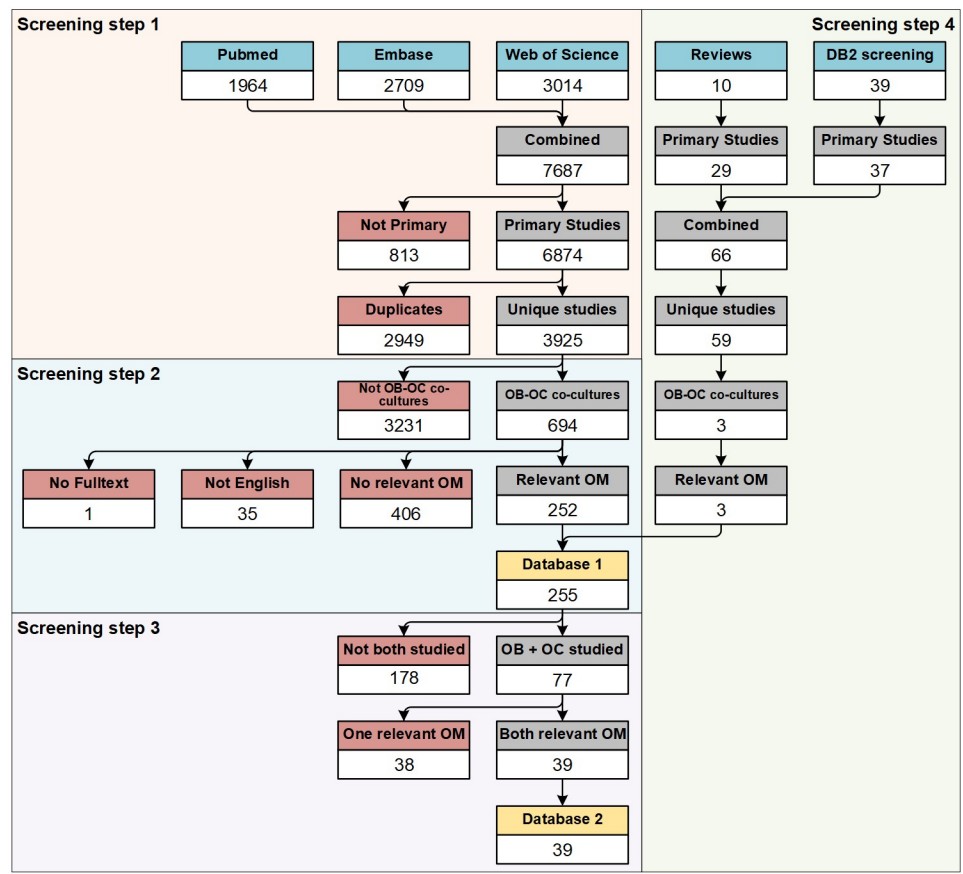

**Fig 1. Flow diagram of systematic literature search and screening.** Screening step 1: Hits from 3 online bibliographic literature sources were combined, primary studies were selected, and duplicates were removed. Title and abstracts were screened for the presence of OB-OC co-cultures. Screening step 2: OB-OC co-cultures were screened in full text for relevant outcome measures. All studies in which at least one relevant outcome measure was studied were included into Database 1. Screening step 3: Papers in which both cell types were studied with relevant outcome measures were included into Database 2. Screening step 4: Papers included into Database 2 and reviews were screened for potentially missing relevant studies and identified studies were screened in the same manner as above. Each screening step is marked with a separate background color. Each selection step within the screening steps is marked with a colored header. Blue header: used as input for the review. Grey header: selection step. Red header: excluded studies. Yellow header: Database as presented in this systematic map. Abbreviations: outcome measures (OM), Database 2 (DB2), osteoblast (OB), osteoclast (OC).

experiments were excluded because these do not allow real-time two-way exchange of cell signals. Explant-, organ- and other *ex vivo* cultures were excluded, except when these were used solely to generate decellularized matrix.

When the study used any type of OB-OC co-culture as defined above, the study was included. When, based on the title and abstract, it was possible that there was a co-culture but this was not described as such, the full-text publication was screened.

**Screening step 2: Identification of relevant outcome measures in the co-culture experiments.** This step was used to identify co-cultures that specifically investigated relevant outcome measures related to bone remodeling: formation or resorption (primary outcome measures), or quantitative measurements of activity markers ALP or TRAP in a dedicated assay (secondary outcome measures). The primary outcome measures of resorption and formation were chosen because these are the processes that are directly affected in bone diseases.

# Systematic literature search

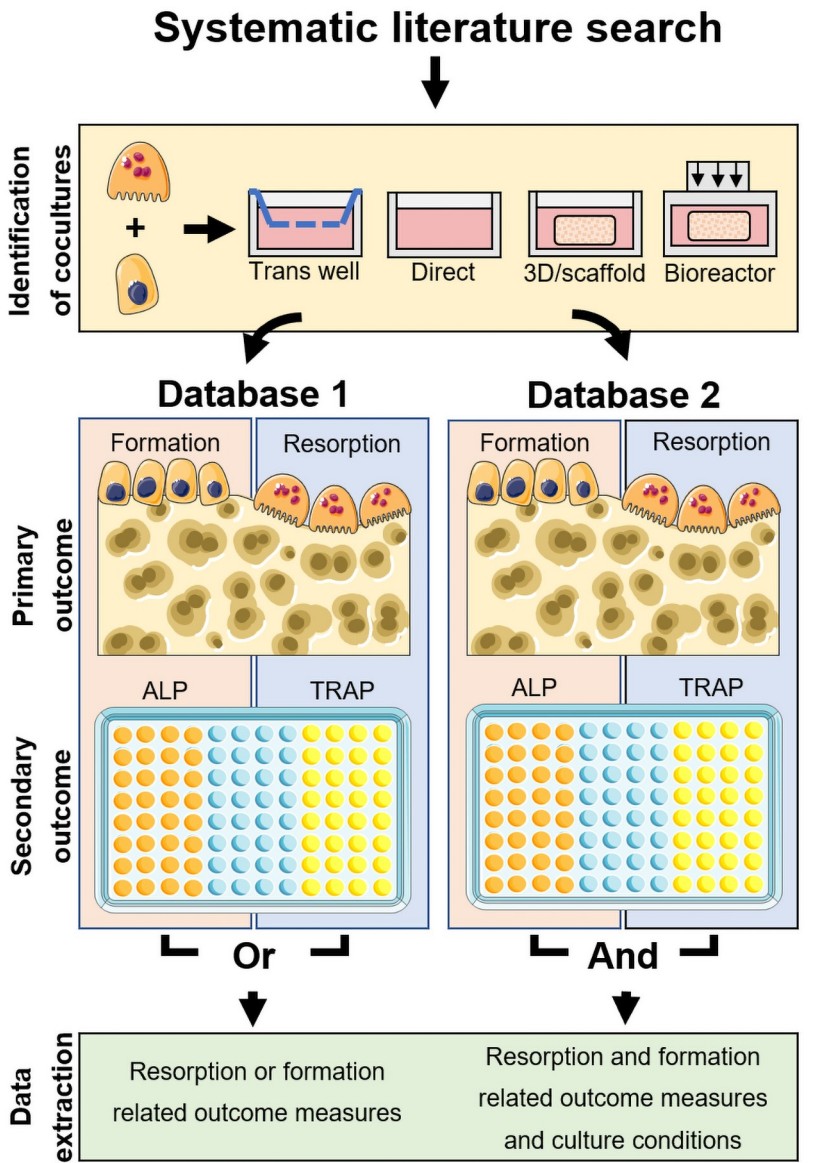

**Fig 2. Schematic overview of Databases 1 and 2.** All identified studies were searched for OB-OC co-cultures, where co-culture was defined as OB and OC being present simultaneously and able to exchange biochemical signals. In addition to direct-contact cultures, cultures such as transwell cultures, 3D or scaffold cultures and bioreactor cultures were allowed as well. OB-OC co-culture studies which used relevant outcome measures were included into Database 1. Of these, only the relevant outcome measures were analyzed. All studies where relevant outcome measures were used for both OB and OC were included into Database 2 as well. Of these, cells and culture conditions were analyzed. The figure was modified from Servier Medical Art, licensed under a Creative Common Attribution 3.0 Generic License (http://smart.servier.com, accessed on 2 July 2021).

Formation/resorption measurement was defined as any method that directly measures the area or volume of (tissue) mineralization by OBs or resorption by OCs or any method that measures by-products or biochemical markers that directly and exclusively correlate to formation/resorption respectively. The secondary outcome measures of ALP and TRAP were included because these are regarded as viable alternatives for the direct measurement of formation and resorption. The measurement of ALP and TRAP was defined as the detection of either

the enzymatic activity or the direct quantification of these proteins present. Polymerase Chain Reaction (PCR) and Immuno-histological stainings (with or without image analysis) were not considered relevant outcome measures. The full texts of the studies identified in screening step 1 were screened for experimental techniques and outcome measures. Studies in which for at least one of the cell types a relevant outcome measure was used were selected to be used in Database 1 (S1 File). Publications written in languages other than English with no translation available and publications where the full text could not be found were excluded at this point.

**Screening step 3: Categorization within Database 1.**   This step made the distinction between studies from screening step 2 on how OBs or OCs were studied in each publication. Each study was categorized into one of five categories within Database 1: 1) A relevant outcome measure was measured in both OBs and OCs in the co-culture. These studies were also included in the in-depth screening for Database 2 (S2 File). 2) and 3) Both cell types were studied, but relevant outcome measures were only measured in OCs or OBs respectively. 4) and 5) Only OCs or OBs respectively were studied in co-culture, the other cell type was neglected.

**Screening step 4: Review and reference list screening.**   To find additional studies that may have been missed during bibliographic searches, relevant review articles and studies labeled as category 1 were screened for additional unique relevant publications. Identified publications were screened as before.

## Database 1 generation and analysis—All co-cultures with relevant outcome measures

All information related to the relevant outcome measures was collected and organized in Database 1. For resorption, additional information on the resorbed substrate, the methodological procedure and quantification of results was collected. For formation, additional information on the type of analysis, the methodological procedure and quantification of results was collected. For both ALP and TRAP, additional information on the mechanism of the biochemical assay, whether it was conducted on lysed cells or supernatant, and information regarding the quantification was collected. In addition, the following information was collected, whether: the authors described their setup as a model specifically for remodeling, the experiment was conducted in 3D, the experiment applied bioreactors, more than 2 cell types were cultured simultaneously, the culture used a trans-well setup, the culture used PCR and components in the supernatant of the culture were analyzed by ELISA or a similar quantification method. Finally, a column for additional remarks was introduced for details that did not fit in another column. Studies where the authors are color coded in pink were those found through screening step 4. Studies categorized as category 1 in screening step 3 were selected for use in Database 2 and had their title color coded in orange.

**Quality assessment and scripting.**   Database 1 only reports the methods used for analyzing relevant outcome measures, and not the data obtained from them or the results described in the publication. Quality assessment in Database 1 is thus limited to assessing the completeness of the necessary elements of the collected methodological details, to the extent that the description of used methods is complete enough to be properly represented in Database 1 and related tables. Publications in which information was missing are here represented as 'not reported' if no information was provided, 'reference only' if no information was provided but another study was referenced, and 'undefined kit', when a commercial kit was used but the content or methodology was not further described. Instances of missing information can easily be identified in figures, tables and databases, but were not further used in this systematic map. Studies where information was missing were still used for other analyses for which the corresponding provided information was present.

A script was written in Excel Visual Basics programming language to analyze Database 1 and extract relevant statistical information on the collected information. On sheet 2 "Data" of the Database 1 excel file, the descriptive statistical data and collected information are presented in the form of lists and tables and together with a button to re-run the analysis based on the reader's requirements. The script is integrated within the excel file and can be used only when the file is saved as a 'macro-enabled' file (.xlsm).

### Database 2 generation and analysis—All co-cultures in which both cell types had relevant outcome measures

Additional information was collected from studies in which relevant outcome measures were studied on both OBs and OCs (Category 1 studies). The species [17], origin (cell line or primary) and cell type [6] of both the OBs and OCs, seeding numbers, densities [18] and ratios [19] were collected or calculated. The culture surface (bio-)material [20], sample size, culture duration, medium refreshing rate, environmental conditions and pre-culture duration [21] were collected if available. The medium components [22] and supplements were extracted, as well as medium components of any monoculture prior to the co-culture. Finally, the tested genes of all studies applying PCR and any proteins studied with ELISA or other supernatant analyses executed on the co-culture were noted.

**Quality assessment and scripting.** In Database 2, the culture conditions, cells and materials used are reported, and not the data obtained from them or the results described in the publication. Quality assessment in Database 2 is thus limited to assessing the completeness of the necessary elements of the collected methodological details, to the extent that the description of used methods is complete enough to be properly represented in Database 2 and related figures and tables. Publications in which information was missing are here represented as 'not reported' (NR) if no information was provided, or 'reference only' if no information was provided but another study was referenced. If studies were missing information critical to reproduce the outcome measures (for example seeding ratio's, culture surface material, medium or supplement information, critical steps in analyses), the cells in the database missing this information were labeled in red. If the missing information was not critical for the outcome measures but necessary for replication of the study (for example sample size, medium refresh rate, control conditions), the cells were labeled in orange.

Three scripts were written using Excel Visual Basics programming language to analyze and process Database 2. One script counts all instances of cells labeled as 'missing info' and present this number in two dedicated columns (missing critical or non-critical info). One script counts the frequency of occurrence of all (co-)authors and years of publication. Finally, one script analyzes this database and extracts relevant descriptive statistical data on the collected information. On sheet 2 "Data" of the Database 2 excel file, the statistical data and collected information are presented in the form of lists and tables together with the buttons to re-run the analyses based on the reader's requirements. The scripts are integrated within the excel file and can be used only when the file is saved as a 'macro-enabled' file (.xlsm).

## Results

### Search results

From three online bibliographic literature sources, 7687 studies were identified (Pubmed: 1964, Embase via OvidSP: 2709, Web of Science: 3014). 6874 studies remained after removing conference abstracts, and 3925 unique studies remained to be screened after duplicate removal.

**Studies included into Database 1 and 2.** After screening step 1, 694 studies remained as OB-OC co-cultures. A list of these studies is available as a (S4 File). Screening step 2 further excluded one study because of missing full text, 35 studies because they were in a language other than English and 406 studies because no relevant outcome measure was used. The qualifying 252 studies were included in Database 1. Screening step 3 revealed that in 77 of the 252 studies in Database 1 both the OB and OC were studied. In 39 of these, both OB and OC were studied using relevant outcome measures. These 39 studies were included in Database 2.

Screening step 4 identified 34 unique studies from the reference lists of the included 39 studies of Database 2, and identified another 25 unique studies from the 10 identified review publications. These additional 59 studies were screened as described previously and resulted in an additional 3 OB-OC co-cultures with only relevant outcome measures measured on one cell type, resulting in a total of 255 studies with relevant outcome measures on at least one cell type for Database 1, and still 39 studies in which relevant outcome measures were studied in both cell types for Database 2. A detailed overview of the search and selection process is shown in Fig 1.

**Publications per year.** The publications included in Database 1 were published between 1983 and 2019, with a peak in publications around the year 2000, followed by a slight but steady increase until now (Fig 3a). The peak roughly coincides with the discovery that M-CSF and RANKL were both necessary and sufficient to induce osteoclastic differentiation in monocytes in 1999 [10]. The included publications in Database 2 span the time between 1997 and 2019, with only 8 publications before 2010 (Fig 3b). This coincides with the progress in development of *in vitro* co-cultures of OBs and OCs, moving beyond co-cultures with OBs to generate OCs, and moving towards co-cultures of OBs and OCs to study for example cell-cell interactions [6].

## Database 1 results

Database 1 provides an overview of all OB-OC co-culture studies published until January 6, 2020 in which at least one relevant outcome measure was studied. Of the 255 studies included, resorption was analyzed in 181 studies, formation was analyzed in 37 studies and both were analyzed in 16 studies. ALP was analyzed in 42 studies, TRAP was analyzed in 61 studies and both were analyzed in 22 studies (Table 1).

**Resorption.** Out of all 255 OB-OC co-culture publications included in Database 1, resorption was studied directly on 188 occasions in 181 studies and quantified 142 times (Table 2a and 2b). In some publications, more than one material or method of analysis for resorption was used. Different materials in the same publication were counted as different studies, resulting in a counted number of studies that is higher than the actual number of publications.

Most studies used discs or fragments of either bone or dentine, visualizing resorption pits directly or after contrast enhancement with stainings. Resorption on bone fragments was quantified using radioimmunological assays measuring the release of *in vivo* pre-labeled $^3$H-proline or type I collagen telopeptide. Synthetic resorbable discs or coatings on culture plates will further be referred to as 'osteologic' plates or discs. Their exact composition is usually not revealed. Resorption of osteologic plates reveal the translucent culture plate, while unresorbed areas are less translucent and can be contrast-enhanced with for example von Kossa's method, facilitating image analysis.

Hydroxyapatite (HA) and other calcium phosphates were used in the form of discs, films, coatings, or scaffolds and were analyzed using various types of microscopy, both with and without prior staining. Resorption of ECM or nodules produced by OBs and scaffolds mineralized by OBs were investigated with transmission electron microscopy (TEM), light microscopy

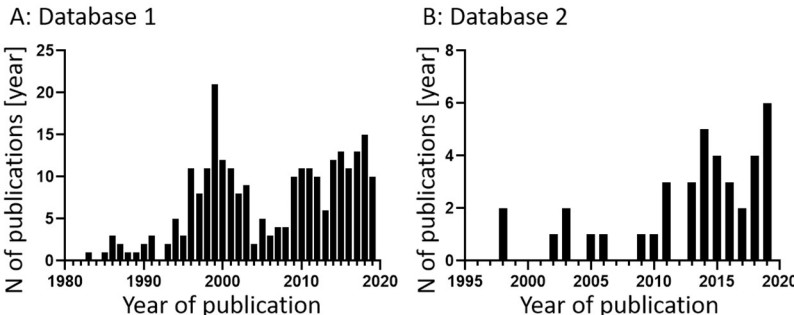

**Fig 3. Relevant publications per year.** A) All 255 publications that contain relevant outcome measures counted by year ranging from 1983 to 2019 (Database 1). B) The 39 selected publications of Database 2 counted by year ranging from 1998 to 2019 (Database 2).

after staining, 2-photon Second Harmonic Generation microscopy [23], supernatant phosphate levels, or with an ELISA for C-terminal telopeptide (CTx) or N-terminal telopeptide (NTx), which are bone turnover marker more commonly used for testing urine and serum samples.

**Formation.** Out of all OB-OC co-cultures included in Database 1, formation was studied directly 39 times in 37 studies and quantified 29 times (Table 3). In some studies, more than one method of measuring, analyzing and quantifying formation was used. In those cases, all methods are counted as individual studies. The methods were divided into 5 types: nodule analysis, volume analysis, surface analysis, supernatant analysis and 3D scans.

The most common method to quantify formation was investigating mineralized nodule formation by staining techniques and/or imaging. Alizarin Red staining could be quantified by releasing the dye from the minerals using acetic acid, followed by spectrophotometry [24]. Surface analysis was to study mineralization on scaffolds, films, or particles. Scaffolds were stained and/or imaged, and the area of matrix deposition was visualized or quantified. Volume analysis was used to describe the measurement of mineralized tissue components calcium and phosphate, which were released after destruction of the matrix. The types of formation measurement above are destructive methods, requiring sacrifice of the samples.

Non-destructive methods include supernatant analysis to describe the measurement of Collagen type I C-terminal propeptide (CICP), a byproduct of collagen deposition, in cell culture

**Table 1. Combinations and frequencies of primary and secondary outcome measures.**

| Combinations of primary and secondary outcome measures in each study | | Primary outcome measures | | | | |
|---|---|---|---|---|---|---|
| | | No resorption or formation | Resorption only | Formation only | Resorption and formation | Total |
| Secondary outcomes measures | No ALP or TRAP | 0 | 151 | 14 | 9 | 174 |
| | ALP only | 16 | 0 | 2 | 2 | 20 |
| | TRAP only | 23 | 9 | 3 | 4 | 39 |
| | ALP + TRAP | 14 | 5 | 2 | 1 | 22 |
| | Total | 53 | 165 | 21 | 16 | 255 |

**Table 1**: This table can be referenced to identify the number of studies using any combination of primary and secondary outcome measures. All 255 studies that investigate at least one of the primary or secondary outcome measures are represented exactly once in this table. Each study is represented by a combination of primary outcome measures (horizontal) and secondary outcome measures (vertical). Marginal totals of each row and column are counted under 'total' with the grand total in the bottom-right cell.

**Table 2.** a. Occurrences of resorption on different types of substrates and subsequent analyses. b. Supernatant resorption techniques.

| | | Materials used as a resorbable substrate for measuring resorption | | | | | | | | | | | |
|---|---|---|---|---|---|---|---|---|---|---|---|---|---|
| **Shapes, structures and types of materials used as resorbable substrate for analysis of resorption.** | | **Dentine** | **Bone** | **HA** | **Silk** | **Collagen** | **CaP** | **PLLA** | **Chitosan** | **Osteologic** | **Mineralized** | **Not reported** | **Per-row total** |
| **Per-material total number of studies** | | 76 | 66 | 6 | 5 | 2 | 4 | 1 | 1 | 19 | 6 | 2 | 188 |
| **Per material quantified studies** | | 55 | 52 | 3 | 4 | 1 | 4 | 0 | 0 | 17 | 4 | 2 | 142 |
| **Shape or structure of material** | **Discs** | 76 | 63 | 2 | | | 2 | | | 13 | | | 156 |
| | **Films** | | | 2 | 4 | | | 1 | 1 | | | | 8 |
| | **Coatings** | | | 2 | | | 1 | | | | | | 3 |
| | **Scaffolds** | | | | 1 | 1 | | | | | 3 | | 5 |
| | **Hydrogels** | | | | | 1 | | | | | | | 1 |
| | **ECM** | | | | | | | | | | 2 | | 2 |
| | **Nodule** | | | | | | | | | | 1 | | 1 |
| | **Fragments** | | 3 | | | | | | | | | | 3 |
| | **Substrates** | | | | | | 1 | | | | | | 1 |
| | **Plates** | | | | | | | | | 6 | | | 6 |
| | **Not reported** | | | | | | | | | | | 2 | 2 |
| **Analysis techniques for analzying resorption on resorbable substrates.** | | **Dentine** | **Bone** | **HA** | **Silk** | **Collagen** | **CaP** | **PLLA** | **Chitosan** | **Osteologic** | **Mineralized** | **Not reported** | **Per-row total** |
| **Staining** | **Toluidine Blue** | 36 | 19 | | | | | | | | | | 55 |
| | **Haematoxylin** | 16 | 2 | | | | | | | | | | 18 |
| | **Eosin** | | 1 | | | | | | | | | | 1 |
| | **H&E** | | 1 | | | | | | | | | | 1 |
| | **Alum / Coomassie Blue** | | 1 | | | | | | | | | | 1 |
| | **TRAP** | 1 | | | | | | | | | | | 1 |
| | **Von Kossa** | | | | | | 2 | | | 4 | 1 | | 7 |
| **Microscopy** | **Phase contrast** | | | | | | 1 | | | 4 | | | 5 |
| | **SEM** | 12 | 37 | 5 | 3 | | 1 | | | 1 | | | 59 |
| | **TEM** | | | | | 1 | | | | | 1 | | 2 |
| | **2-Photon** | | | | | | | | | | 1 | | 1 |
| | **Atomic force** | | | | 1 | | | 1 | 1 | | | | 3 |
| | **Reflected light** | 8 | 2 | | | | | | | | | | 10 |
| | **Dark field** | | | | | | | | | 1 | | | 1 |
| | **Light microscopy** | | | | | | | | | 6 | | | 6 |
| **Other** | **Assay** | | | | | 1 | | | | | | | 1 |
| | **Immuno-assay** | | 3 | | | | | | | | 3 | 1 | 7 |
| | **MicroCT** | | | | 1 | | | | | 1 | | | 2 |
| | **Reference only** | 2 | | | | | | | | | | | 2 |
| | **Not reported** | 1 | | 1 | | | | | | 2 | | 1 | 5 |
| | **Total per material** | 76 | 66 | 6 | 5 | 2 | 4 | 1 | 1 | 19 | 6 | 2 | 188 |
| **Supernatant Analysis techniques per material used for analysis of resorption.** | | **Dentine** | **Bone** | **HA** | **Silk** | **Collagen** | **CaP** | **PLLA** | **Chitosan** | **Osteologic** | **Mineralized** | **Not reported** | **Per-row total** |

(*Continued*)

**Table 2.** (Continued)

| | | Materials used as a resorbable substrate for measuring resorption | | | | | | | | | | Total |
|---|---|---|---|---|---|---|---|---|---|---|---|---|
| Supernatant analysis | NTx | 1 | | | | | | | | | 2 | | 3 |
| | CTx | | | | | | | | | | 1 | 1 | 2 |
| | ICTP | | 1 | | | | | | | | | | 1 |
| | Phosphate release | | | | 1 | | | | | | 2 | | 3 |
| | Radioactive proline release | | 2 | | | | | | | | | | 2 |

Table 2a: Each column signifies a different material used as a substrate for measuring resorption. The first rows show how many instances of each material were included into this systematic map in total, and how many times the results were quantified. The final column shows incremental totals per material type or analysis type. This table consists of two sections. The top section shows in what form or shape the corresponding materials were used as a substrate for resorption. The bottom section shows the techniques that were used to study the resorption described on the materials described in the top section. Each individual study is represented exactly once in the top section of the table to signify the type and form of the substrate used, and exactly once in the bottom section of the table to signify the method used to analyze the resorption that occurred on these substrates. This required the selection of the most 'important' part of the methods used. In the cases where first a staining was used followed by microscopy, only the staining is listed. Only in those cases where resorption was investigated directly with a microscope without prior staining, the type of microscopy is listed. 'Mineralized' = *A priori* mineralized matrix by other cells.

Table 2b: This table presents five resorption analyses that can be measured in the culture supernatant and not on the material itself. They are presented separately because they were done in addition to 'regular' analyses (Table 2a).

**Table 3. Occurrences of different methods of formation detection and subsequent analyses.**

| | | Type of analysis used to measure formation | | | | | |
|---|---|---|---|---|---|---|---|
| | Technique | Scan | Nodule analysis | Supernatant analysis | Surface analysis | Volume analysis | Per-row Total |
| | Total | 3 | 20 | 6 | 5 | 5 | 39 |
| | Quantified | 3 | 12 | 6 | 3 | 5 | 29 |
| Measured shape or structure | Scaffold | 2 | 1 | 1 | 3 | 2 | 9 |
| | Film | 1 | | 1 | 1 | 2 | 5 |
| | Hydrogel | | | | | 1 | 1 |
| | Pellet | | 1 | | | 1 | 2 |
| | Dye release | | 5 | | | | 5 |
| | Analysis | Scan | Nodule analysis | Supernatant | Surface analysis | Volume analysis | Per-row Total |
| Staining | H&E | | | | 1 | | 1 |
| | Von Kossa | | 2 | | 1 | | 3 |
| | Alizarin Red | | 16 | | | | 16 |
| | Lentiviral fluorescence | | 1 | | | | 1 |
| Assays | Calcium | | | | | 3 | 3 |
| | Calcium + Phosphate | | | | | 2 | 2 |
| | CICP | | | 6 | | | 6 |
| Other | SEM | | 1 | | 3 | | 4 |
| | microCT | 3 | | | | | 3 |
| Per-analysis Total | | 3 | 20 | 6 | 5 | 5 | 39 |

Table 3: Each column signifies a different type of analysis used for measuring formation. The first rows show how many instances of each type of analysis were included into this systematic map in total, and how many times the results were quantified. The final column shows marginal totals per row of each row. This table consists of two distinct sections, each starting with a row showing all analysis types for convenience. The first section lists defining characteristics of studies such as using films, scaffolds, hydrogels or pellets, or using a technique to first stain tissue, and then releasing and measuring the released dye. Not each study had such defining characteristics, and the total of section one does not add up to 39 studies. Section two shows either which materials was measured, or which technique was used for measuring formation. Each instance of formation is represented in section two of this table exactly once.

supernatant. 3D scanning by μCT quantified the three-dimensional structure of mineralized matrix.

**TRAP measurements as a surrogate marker of osteoclastic resorption.** Out of all OB-OC studies in Database 1, the predominant OC marker TRAP [25] was studied 63 times in 61 publications (Table 4). TRAP can be measured intracellularly or excreted into the medium, either by measuring its enzymatic phosphatase activity directly, or by quantifying the amount of TRAP molecules present. TRAP release was studied both on cell lysate and on supernatant, and in some cases on both. The most frequently used method to study TRAP activity was using 4-nithophenylphosphate (pNPP). Others used the fluorophore Naphthol ASBI-phosphate [26] which shows specificity for TRAP isoform 5b [27]. Naphtol ASMX phosphate [28] and an otherwise undisclosed diazonium salt function in a similar manner. Enzyme linked Immunosorbent Assay (ELISA) was used to detect TRAP using conjugated enzymes or fluorophores,. Others used a kit to detect TRAP, but no description of the assay other than the manufacturer were given.

**ALP measurements as a surrogate marker of osteoblastic tissue formation.** Bone turnover marker ALP was studied in 42 publications (Table 5). ALP was most frequently measured using pNPP as substrate which is converted by ALP itself. Enzyme Immuno Assays (EIA) and ELISAs are immunoenzymatic assays [29] that label ALP molecules with a detectable substrate or other enzymes. Others used a kit to measure ALP, but no description of the assay other than the manufacturer were given.

## Database 2 results

While Database 1 provides an overview of all reported methods to study the relevant outcome measures (resorption, formation, TRAP and ALP), Database 2 provides more experimental details such as culture conditions used for co-cultures.

**Osteoblasts.** Database 2 included 39 studies. Table 6 presents the cell types at the start of the co-culture (Table 6). Most studies used human primary cells. Almost half of the studies started the co-culture with OBs, the others started with progenitor cells. As a result of ambiguous isolation methods and nomenclature which is subjective and can evolve over time [30], some cell descriptions in Table 6 might refer to identical cell populations. This systematic map reflects the nomenclature used by the authors or extrapolated from the description and does not further interpret the provided information.

Except for the oldest 6 studies that used chicken and rat cells, all studies used human or mouse cells, most of which were primary cells. While the studies using rat and mouse cells mostly directly introduced OBs (either isolated as such or differentiated before seeding), those that used human cells predominantly introduced progenitor cells [30]. Those that used primary OBs purchased expandable human OBs [31] or used OBs [32], undefined expanded bone cells [33], or differentiated MSCs [34] from bones obtained during a surgical procedure.

OB Seeding densities ranged from $0.9 \times 10^3$ cells/cm$^2$ to $60 \times 10^3$ cells/cm$^2$ with a mean of $11 \times 10^3$ cells/cm$^2$ (N = 26) in 2D (Fig 4a) and from $0.3 \times 10^3$ cells/cm$^3$ to $7 \times 10^3$ cells/cm$^7$ with a mean of $15 \times 10^6$ cells/cm$^3$ (N = 6) in 3D (Fig 4d).

**Osteoclasts.** Out of the 39 studies in Database 2, 20 used human primary cells, the others used animal primary cells or any type of cell line for resorption (Table 7). Cultures were mostly initiated with OC progenitors: 16 studies introduced monocytes, 11 introduced mononuclear cells, the rest used other precursors.

The 6 oldest included studies used chicken and rat cells, all others used mouse or human cells. With only one exception combining a mouse ST-2 cell line with human monocytes [35], all studies used cells of exclusively a single species for the OB and OC source. Only one study

**Table 4. TRAP measurement techniques and analyses.**

| Type | pNPP | N-ASBI-P | N-ASMX-P | ELISA | Diazonium salt | Undefined kit | Reference | Not reported | Total |
|---|---|---|---|---|---|---|---|---|---|
| Total | 33 | 5 | 1 | 9 | 1 | 9 | 4 | 1 | 63 |
| Lysed cells | 29 | 5 | 1 | 1 | | 3 | 2 | | 41 |
| Supernatant | 6 | | | 7 | 1 | 6 | 2 | | 22 |
| Reference only | | | | 1 | | | | | 1 |
| Not reported | | | | | | 1 | | 1 | 2 |
| Analysis | pNPP | N-ASBI-P | N-ASMX-P | ELISA | Diazonium salt | Kit | Reference | Not reported | Total |
| absorbance | 33 | | 1 | 8 | | 6 | 2 | 1 | 51 |
| Fluorescence | | 5 | | | | | | | 5 |
| Reference only | 2 | | | 1 | 1 | | 2 | | 6 |
| Not reported | | | | | | 4 | | | 4 |

**Table 4**: Each column in Table 4 signifies a different technique to measure TRAP. This table consists of two distinct sections. The first section shows the number of studies that used each technique, and whether these were used on (lysed) cells or on culture supernatant. The second section shows with which method of analysis the TRAP content was analyzed. If one study measured TRAP on both cells and supernatant, then that study is represented twice in both sections resulting in a higher count of occurrences than number of studies that analyzed TRAP. In all other cases, each study is represented once in each section.

claimed to introduce OCs directly into co-culture but failed to provide any information regarding the cell source and was therefore ignored from further investigation.

The OC seeding density ranged from $5 \times 10^3$ cells/cm$^2$ to $15 \times 10^6$ cells/cm$^2$ with a mean of $190 \times 10^3$ cells/cm$^2$ (N = 25) in 2D (Fig 4b) and from $20 \times 10^3$ cells/cm$^3$ to $70^*10^6$ cells/cm$^3$ with a mean of $17 \times 10^6$ cells/cm$^3$ (N = 6) in 3D (Fig 4e). Seeding ratios of OB:OC in 2D varied highly and ranged from 1:1500 to 1:1 (Fig 4c). seeding ratios of OB:OC in 3D ranged from 100:1 to 1:25 (Fig 4f).

**Co-culture medium composition and culture conditions.** The behavior of cells is highly dependent on their environment, of which the biochemical part is predominantly determined by the culture medium composition. The main components of typical culture media are a base medium, fetal bovine serum (FBS) and specific supplements such as OB and OC supplements. 8 different base (or complete) media were reported (Fig 5a), with αMEM and DMEM accounting for approximately 80% of all studies. FBS content ranged from 0% to 20%, with most studies using 10% (Fig 5b). Those without supplemented FBS used forms of complete media of which the composition was not described, but possibly including a type of serum or equivalent serum-free supplements.

**Table 5. ALP measurement techniques and analysis.**

| | | ALP measurement techniques | | | | |
|---|---|---|---|---|---|---|
| | Type | pNPP | EIA | ELISA | Undefined kit | Total |
| Substrate | Total | 26 | 8 | 1 | 7 | 42 |
| | Lysed cells | 19 | 1 | | 6 | 26 |
| | supernatant | 8 | 7 | 1 | 2 | 18 |
| Detection | absorbance | 25 | 8 | 1 | 3 | 37 |
| | Reference only | 2 | | | | 2 |
| | Not reported | | | | 5 | 5 |

**Table 5**: Each column signifies a different technique to measure ALP. The first rows show the occurrence of each technique and whether these were used on (lysed) cells, or on culture supernatant. The final three rows show with which method of analysis the ALP content was measured. In a single study ALP can be measured with the same technique on both cell lysate and culture supernatant, resulting in a higher count of occurrences than number of studies that analyzed ALP.

**Table 6. Osteoblast origins and occurrences.**

| Cell Origin | Osteoblasts | Mesenchymal stem cells | Mesenchymal stromal cells | Stromal cells | Stromal vascular Fraction | Osteoprogenitor cells | Per-row Total |
|---|---|---|---|---|---|---|---|
| Human primary | 4 | 9 | 2 | 6 | 1 | | 22 |
| Human cell line | 1 | | | | | | 1 |
| Mouse primary | 3 | 2 | | | | | 5 |
| Mouse cell line | 4 | | | | | | 4 |
| Rat primary | 3 | | | | | 1 | 4 |
| Chicken primary | | | | 2 | | | 2 |
| Reference only | 1 | | | | | | 1 |
| Total | 16 | 11 | 2 | 8 | 1 | 1 | 39 |

**Table 6**: From Database 2, the origin of the cells that were used as OB was extracted. Each column represents a different cell type of OB-like cells or their precursors. Each row represents a different source of cells, differentiating between both the origin species and whether the cells are primary cells or cell lines. Incremental totals are presented in the last row and column.

M-CSF concentration was reported in 11 studies and ranged from 10 ng/ml to 100 ng/ml with a mean of 39,82 ng/ml (Fig 5c). RANKL concentration was reported in 14 studies and ranged from 10 ng/ml to 100 ng/ml with a mean of 49 ng/ml. OB supplements were recalculated to molarity if necessary (Fig 5d). Ascorbic Acid (AA) (also referred to as ascorbic acid-2-phosphate, L-ascorbic acid or L-ascorbate-2-phosphate) concentration was reported in 19 studies and ranged from 0.05 mM to 0.57 mM, with a mean of 0.18 mM and one outlier at 200 mM that was disregarded for this calculation. Dexamethasone was used in 13 studies and was used in 2 different molarities: 6 times at $10^{-7}$ M and 7 times at $10^{-8}$ M. β-Glycerophosphate

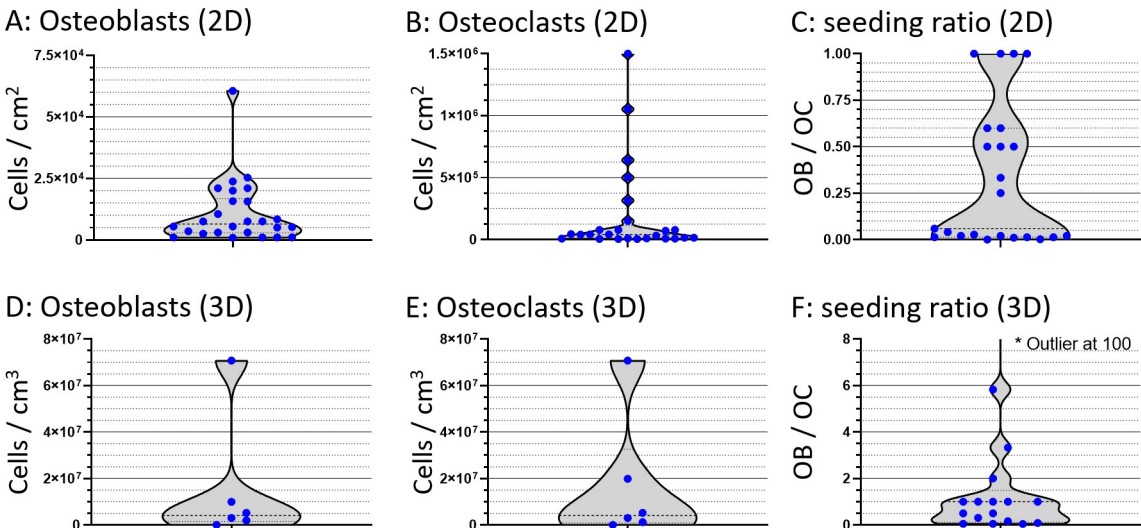

**Fig 4. Seeding densities and seeding ratios.** Violin plots of 2D and 3D seeding ratios of OB (A+D), OC (B+E) and respective seeding ratios in co-cultures (C+F). Values are calculated based on reported seeding numbers of the cells or precursors thereof per surface are or volume. No distinction was made between different (precursor) cell types in these figures, resulting in a considerable spread in data that could be attributed to proliferation and cell fusion after seeding The ranges along the Y-axis are not the same for each figure. Each seeding density of each study is represented by a blue dot.

**Table 7. Osteoclast origins and occurrences.**

| Cell Origin | Monocytes | Mononuclear cells | Macrophages | Osteoclast precursors | Osteoclasts | Spleen cells | Total |
|---|---|---|---|---|---|---|---|
| Human primary | 10 | 6 | 1 | 3 | | | 20 |
| Human cell line | 4 | | | | | | 4 |
| Mouse primary | 2 | | 2 | 2 | | | 6 |
| Mouse cell line | | | 2 | | | | 2 |
| Rat primary | | 3 | | | | 1 | 4 |
| Chicken primary | | 2 | | | | | 2 |
| Reference only | | | | | 1 | | 1 |
| Total | 16 | 11 | 5 | 5 | 1 | 1 | 39 |

**Table 7**: From Database 2, the origin of cells used as OC was extracted. Each column represents a different cell type of OC-like cell or a precursor. Each row represents a different source of cells, differentiating between both the origin species and whether the cells are primary cells or cell lines. Incremental totals are presented in the last row and column.

(βGP) concentration was reported in 17 studies, and ranged from 1 mM to 46 mM, with a mean of 13 mM.

## Discussion

In recent years, many research groups have ventured into the realm of OB-OC co-cultures with the intent of studying both formation and resorption. Due to a lack of standardization within the field and the difficulty of finding publications based on methods instead of results,

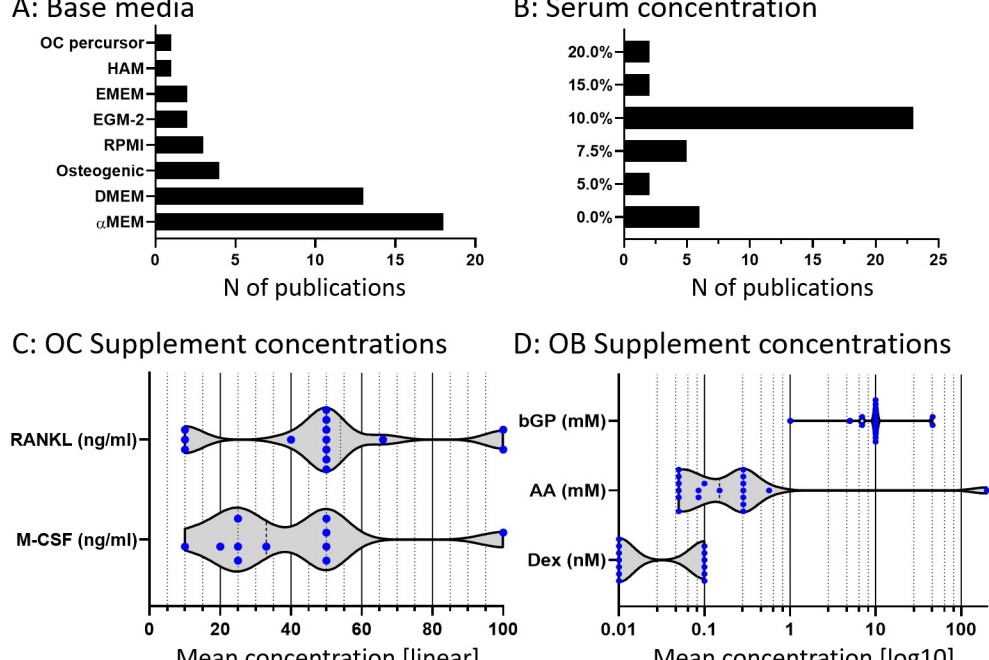

**Fig 5. Medium components used by studies in Database 2.** A) The occurrence of all identified base and complete media used during the co-culture phase of each study. B) Serum concentrations during the co-culture phase of each study. C) OC supplements administered during the co-culture phase of each study. Please note that the x-axis has a linear distribution. D) Osteogenic supplements during the co-culture phase of each study. Please note that the x-axis has a logarithmic scale. Individual concentrations or molarities are shown as blue dots.

each group seems to be individually developing the tools to suit their needs resulting in many functionally related experiments that are methodologically different. The use of OB-OC co-cultures is usually not clearly mentioned in the title and abstract, making it difficult to find these studies without a systematic search and thorough review. The aim of this study was to generate a systematic map to give an overview of existing osteoblast-osteoclast co-culture studies published up to 6 January 2020, and present their methods, predetermined outcome measures and other useful parameters for analysis in 2 databases which can be filtered, sorted, searched and expanded.

The Database 1 contains all OB-OC co-culture studies in which at least one relevant primary outcome measure (formation and/or resorption) or secondary outcome measure (ALP and/or TRAP quantification) was investigated (S1 File). A sub-selection of studies that have relevant outcome measures investigated on both OBs and OCs in the co-culture are shown in Database 2, accompanied by additional details on methods, culture conditions and cells (S2 File).

## Resorption

Most studies in Database 1 investigating resorption did so in 2D cultures using a resorbable substrate such as bone, dentine, or synthetic osteological discs. This is not unexpected, as these three options are either the actual *in vivo* material (bone), a similar material with excellent properties for studying resorption (dentine) [36], or a material designed specifically for the purpose of studying resorption (osteologic discs or coated wells). Dentine is a component of ivory, usually obtained from elephants [37], hippo's [38] or sperm whales [39]. Regulations regarding ivory are strict and the material is rare, making it difficult to obtain. One crucial advantage of using dentine over bone is related to the native structure of dentine itself: it does not contain canaliculi and has fewer other irregularities, providing more contrast between the native structure and resorption pits to accurately visualize them [36, 40]. The advantages of bone over dentine are that bone is the actual tissue of interest, it can be obtained from many different species in relevant quantities and sizes, it can be prelabeled *in vivo* with radioactive markers such as $^3$H-proline [41], and could be used in conjunction with cells from the same species or even same animal, although the latter was not observed in this map. Synthetic osteologic discs have the advantage of being produced in a uniform manner and should show little sample-to-sample variation compared to discs made from animal tissue or hand-made discs. Using well plates with thin osteologic coatings has the advantage that once the coating is resorbed the translucent well below is revealed, which facilitates imaging with light microscopes. Combined with certain stainings, it makes quantifying resorbed area using conventional light microscopy easier.

It is believed that the deposition of collagen type I by osteoblasts is a vital step in the formation of mineralized tissue [42], and similarly could play a role in the resorption thereof. When using collagen-based materials, techniques such as NTx [43] and CTx [44] can be used. These bone turnover markers are used in the clinic and can quantify resorption by analyzing the liberated collagen fragments in the supernatant [45]. It is possible to generate the to-be-resorbed material *in vitro* by OBs [44], even within the same experiment. This simulates a bone remodeling environment that is a step closer to the physiological process of bone remodeling versus only resorption, although *in vivo* the order is typically reversed: first, ECM is resorbed by OC, then new ECM is deposited by OB [46]. However, the process of creating a mineralized matrix may introduce a variation in substrate size even prior to initiating the co-culture [47].

Because most studies were conducted in 2D, most resorted to using various types of 2D microscopy to analyze resorption, usually after staining to increase contrast. This can facilitate

the quantification of resorbed area using image analysis software but is usually limited to a quantification of surface area, whereas resorption is a three-dimensional process. While methods exist to reconstruct a set of stereoscopic 2D images into 3D height maps [48], these were not identified within the studies in either database of this systematic map. Instead, one could also consider techniques that can directly quantify the resorbed volume. Examples are 2-photon microscopy for thin samples and micro computed tomography (μCT) [49]. Due to the non-destructive nature of μCT, it is well suited to monitor mineralized volume change over time within the same samples over a longer period of time [47, 49, 50]. Registering consecutive images can even show both formation and resorption events within the same set of images of the same sample if both mineralizing OBs and resorbing OCs were present [47]. The usefulness of such a monitoring tool is however dependent on the envisaged resolution versus the corresponding potential cell-damage caused by radiation exposure [51, 52], and requires the use of sterile scannable culture vessels, which poses some practical constraints. While μCT in this map is predominantly used on 3D samples, one study used it to quantify the thickness of mineralized films and combined that data with a surface metrological analysis [53].

Overall, the golden standard (bone and dentine discs) remains the most-used method to study 2D resorption, although alternatives such as osteological coatings offer new and easy ways of quantification. Compared to 2D cultures however, 3D cultures are under-represented in this systematic map. Only 24 studies were labeled as 3D co-cultures in Database 1, the first being published only in 2006 [35]. From these we learn that studying 3D resorption remains a challenge, with the only identified viable options for quantification being μCT imaging and supernatant analysis techniques such as NTx and CTx.

## Formation

Bone formation is a multi-step process in which properly stimulated OBs lay down a framework of type I collagen, which in turn is mineralized with calcium phosphate [42]. No single method of measuring formation confirms the occurrence of each step in this process, instead relying on the assumption that the confirmed presence of one step indicates the presence of the entire process.

With most studies being 2D co-cultures, it is no surprise that most formation analyses were stainings. Of these, Alizarin Red is particularly interesting due to the possibility of quantifying the amount of bound dye, which correlates to the amount of calcium [24]. A risk when using this method on larger samples is that it is not certain how far both dye application and dye extraction penetrate the material. This should not affect relative comparisons between different sample groups but could lead to underestimations of calcium deposition. By completely lysing the samples and directly measuring the exact amount of calcium or phosphate [54, 55] this risk could be avoided, at the cost of not gaining information on the distribution of calcium or phosphate through the sample.

The two types of non-destructive formation measurements, CICP and μCT, are coincidently well-suited for the analysis of three-dimensional co-cultures. Because of their non-destructive nature, they can be used to measure the same sample repeatedly and prior to destructive techniques. CICP measurements [56] have no negative effects on the co-culture, requiring only a culture supernatant sample. The use of μCT leads to both quantification and visualization of mineralization within the same sample over time, but needs some consideration because the same constraints described for resorption apply here as well.

Overall, 2D nodule stainings were the most frequently used method to measure formation. Combined with Alizarin Red dye release these provide an easy way to quantify mineralization,

though CICP supernatant analysis and μCT techniques provide a non-destructive alternative that can also be used for 3D co-cultures.

## ALP and TRAP

ALP and TRAP are the two major markers for indirectly quantifying OB and OC activity that were included into Database 1. ALP makes phosphates available to be incorporated into the matrix [57] and TRAP has been associated with migration and activation of OC [58]. Their presence is not conclusive proof that formation and resorption are occurring because ALP is expressed already in differentiating MSCs [59] and TRAP is expressed on monocytes as well [47]. Still, there is a correlation between their presence and that of OB and OC activity. These enzymes can be measured both after lysis of the cells or within the culture supernatant. The former allows the quantification of enzyme per DNA content when combined with a DNA assay, whereas the latter allows the monitoring of relative enzyme release over time. The most frequently used methods are the pNPP-based methods where ALP and TRAP directly convert a substrate into a measurable compound. Naphthol-based methods [26] rely on a similar principle, and show an increased specificity for TRAP isoform 5B in particular [27]. The main advantage of these methods is that they use the inherent enzymatic activity of ALP and TRAP, reducing the complexity and cost of the assay. However, the reliance on the inherent enzymatic activity of the enzymes is also a practical limitation as inherent activity can be affected by for example freeze-thaw cycles and long-term storage, which is a likely occurence when monitoring ALP or TRAP release over time. A workaround would be to analyze the samples directly after collection. Another risk is that both ALP and TRAP are phosphatases. Assays that rely on their inherent phosphatase activity may show cross-reactivity of other phosphatases, although this can largely be mitigated by controlling the pH during the test.

Immunoenzymatic assays such as ELISA [60] detect the presence and not the activity of these enzymes instead. These methods have the capacity to detect low protein concentrations because each individual protein can be labeled with an excess of new enzymes each capable of converting substrate. In the case of TRAP, ELISA kits exist that are specific for TRAP isoform 5b which is expressed almost exclusively in OCs [61], whereas isoform 5a is also expressed by macrophages and dendritic cells [62]. While in a co-culture with pure populations of OB and OC this distinction would not be relevant, macrophages, macrophage-like cells and macrophage precursors [63] can be used as precursors for OCs [23], and thus express isoform 5a in co-culture. Whether this negatively affects the results is another matter that can only be determined by comparison between the two assay types.

To conclude, pNPP based methods are the most frequently used methods for detecting ALP and TRAP due to their affordability and simplicity. However, immunoenzymatic detection methods are more sensitive and specific, and do not rely on the intrinsic enzymatic activity of ALP and TRAP which can be affected by freeze-thaw cycles, long-term storage, and could show cross-reactivity with other phosphatases.

## Osteoclasts

Osteoclastic resorption is an integral part of *in vivo* bone maintenance. Old and damaged bone tissue is resorbed and replaced by OBs with new bone tissue. There is a clear preference in the studies identified for Database 2 for using human cells to generate OCs, most notably monocytes and mononuclear cells. These have in the past two decades proven to be a reliable and relatively straight-forward precursor population for OCs [6], they can be obtained from human blood donations, and are thought to be better representatives for studying human physiology than cells of animal origin [2, 3].

The choice of using precursors versus differentiated OCs is forced sharply into one direction because of both biological and experimental limitations. The extraction of OCs from bone is possible but cumbersome, requires access to fresh bone material and generally does not yield relevant numbers of OCs. Generating OCs from circulating precursors is easier. However, OCs have an average life span of approximately 2 weeks [64, 65], some of which would already be lost if OCs would be created prior to the actual experiments. In contrast to most cells, differentiation happens by fusion of several precursors into a single OC. Fused multinucleated OCs can become large and hard to handle without being damaged. For those reasons they are usually differentiated from precursors within the actual experiments.

Thanks to the discovery of M-CSF and RANKL being sufficient to induce osteoclastic differentiation [10], OCs can currently be obtained *in vitro* without the need for OBs. Where in the past researchers used spleen cells for this, the studies included in this systematic map predominantly use (blood-derived) mononuclear cells, monocytes, or macrophages as precursor cells.

There are *caveats* and risks associated with each cell source. Animal cells introduce a between-species variation and can respond differently than human cells [17]. Human donor cells tend to exhibit large between-donor variation compared to cell lines [66] and the number of cells acquired is limited and variable [67]. The large variation between donors again highlights the need for patient-specific disease models instead of generic bone models. By using cells of a single diseased donor, the reaction of that patient's cells on potential treatment options can be studied. Immortalized cell-lines are more practical than primary cells but result in immortal OC-like cells. While these can greatly reduce between-experiment and between-lab variation, they are also physiologically less relevant. While these risks and characteristics do not discredit any source as a viable source of OCs for any experiment, the results of the corresponding studies should be interpreted with these characteristics in mind.

## Osteoblasts

OBs are the bone forming cells, and together with bone resorbing OCs they keep the bone mass and bone strength in equilibrium. The preference for the use of human primary cells identified in the studies included in Database 2 can be explained by the good availability of donor material, expandability of OB precursors, and because human cells better reflect human physiology than cells from other species [2, 3]. The choice of OB progenitors versus OBs is not as crucial here as it is with OCs. MSCs, the most commonly used precursors, have a tri-lineage potential [68] and differentiate into OBs on a 1–1 ratio. The advantage of osteoprogenitors such as MSCs is that these are capable of extensive proliferation before differentiation. Using progenitors allows studying osteoblastogenesis in addition to bone formation. When the effect of an intervention on mineralization but not osteogenesis is under investigation, care must be taken that the intervention is not applied before differentiation has been achieved.

The advantage of directly introducing OBs instead of precursors, whether obtained directly from primary material or pre-differentiated *in vitro*, is that these do not need to be differentiated within the experiment anymore, and any experimental conditions affect only mature OBs and not osteoblastogenesis in parallel. OBs or to-be-differentiated MSCs isolated from bone marrow or orthopedic surgery are the most common source of primary human OBs. Healthy human donor OBs are scarce because these persons rarely undergo bone surgeries or get bone biopsies. Whether the use of OBs from diseased donors affects experimental results needs to be elucidated. On the other hand, using patient cells to create a personalized *in vitro* disease model is the first step towards personalized medicine, especially if all cells are of that same patient. Finally, the risks of using animal cells the introduction of a between-species variation.

While none of these risks directly discredit any of the methods obtaining OBs, the results must be interpreted with these risks and characteristics in mind.

## Culture conditions

The success of a cell-culture experiment is dependent on the culturing conditions. For many cell-types, optimal culture conditions have been established. During co-culture experiments however, the needs of two or more cell types need to be met. Medium components and factors may be needed in different concentrations, as they can be beneficial to one cell type but inhibitory to the other [69].

There is a clear preference for medium based on DMEM and αMEM, but many factors influence the choice of base medium. Base media are chosen based on the intended cell type, recommendations by a manufacturer or supplier of either cells or medium, preferred effect on cells, interaction with other supplements, and earlier experience. These factors make direct comparison of experimental results within literature virtually impossible. Additionally, none of the studies mentioned why they specifically chose the base media they used.

Another variable in medium composition is FBS (or FCS). It is known to have batch-to-batch- and between-brand differences [70] which can impact the results of an experiment tremendously. However, no study explains why each type and concentration of FBS was chosen.

When osteoblastic or osteoclastic supplements were used, the concentrations were within the same orders of magnitude in all studies, except for AA. Only 2 studies used all 5 of the supplements indexed in this study (AA, βGP, Dexamethasone, M-CSF and RANKL) and many combinations of supplements have been registered in this map. OC supplements RANKL and M-CSF are both necessary and sufficient for osteoclastogenesis [10]. However, OBs can produce RANKL and M-CSF themselves to trigger differentiation [9] and therefore the supplements are not necessarily required in co-culture. Each osteoblastic supplement contributes to a specific function. Dexamethasone upregulates osteogenic differentiation, βGP acts as a phosphate source, and AA is a co-factor involved in collagen synthesis [71]. Depending on the type of (progenitor) cells introduced, the aim of the experiment and other methodological details, their inclusion could be necessary. Finally, many studies used or omitted specific supplements related to their research question regarding the activity of OBs or OCs or used less common supplements for differentiation such as vitamin D3, human serum or Phorbol 12-myristate 13-acetate.

What is seldom addressed however, is the compromise that must be made in choosing the right supplements and concentrations. Adding too high doses of supplements could cause an excess of these signals in the culture medium, effectively overshadowing any other ongoing cell-signaling over the same pathway by other cells. This is of critical importance when the goal is not to achieve only OB and/or OC activity, but a self-regulating system with experimental conditions or interventions that are expected to affect this system. Here, it may be beneficial to experiment with lower concentrations of factors, supplemented only during critical phases of the cells' development or differentiation.

The choice of medium in a co-culture is most likely going to be a compromise and must be based on the exact research question to be addressed, where the advantages and disadvantages of base media and supplements for both cell types are carefully weighed.

**Seeding densities and seeding ratios.** Using the correct seeding densities plays a major role in proliferation and cell function of OBs [18, 72] and osteoclastic differentiation [73]. The seeding densities reported in this map show an enormous spread. Many factors could have influenced these numbers. For example, some studies report the numbers prior to expansion, others expand the cells in (co-)culture. Similarly, the percentages of relevant precursor cells in

heterogenous cell populations can vary widely. The cell numbers present and OB:OC ratio most likely even change during a co-culture due to ongoing cell-division, differentiation, fusion and different expected life spans and corresponding cell death. Regrettably, the available documentation of exact cell numbers introduced is often lacking, and open to some interpretation.

Animal type, cell type, cell line versus primary cells and even passage number may also directly influence the choice of seeding densities in addition to various experimental choices. At the same time, the purpose of the experiment and more specifically the purpose of the cells and type of interaction or result required should determine the necessary seeding density. The combination of all these factors suggests that there in fact is no ideal seeding density, and that the best seeding density for a certain experiment can only be determined by taking all the above factors into account, learning from others that did similar experiments, and most importantly verifying assumptions and predictions in the lab.

Looking at the cell seeding ratio, here reported as number of seeded OB/OB-precursors per seeded OC/OC-precursor, outliers can be normalized against their seeded counterparts. In 2D studies, there are never more OBs/OB-precursors than OCs/OC-precursors. At most, they are seeded at a 1:1 OB:OC ratio. Even though in human bone tissue the ratio of OB:OC is estimated to be approximately 7:1 [74], higher OC numbers than OB numbers are seen. OB precursors can still proliferate, whereas OC precursors usually still need to fuse together to form mature OC or OC-like cells. In 3D we do not see the same trend, with ratio's ranging from 1:20 to 100:1. These differences are again affected by the same factors that influence individual OB and OC seeding densities, further enhanced by the extra layer of complexity that are inherent to 3D cultures. As with the individual seeding densities, these factors prevent us from determining an ideal seeding ratio.

## Limitations

While the authors took great care to construct a series of search queries fine-tuned for each of the three online bibliographic literature sources, the authors cannot be certain that all relevant OB-OC co-cultures have been included into the two databases. The search was limited by the necessary addition of a 'co-culture' search element. Co-culture studies without any indication thereof in the title or abstract simply cannot be identified through the initial search. To compensate for this, screening step 4, searching through identified reviews and publications included into Database 2, was executed. Publications in languages other than English were excluded because none of the researchers involved in data curation and analysis were fluent in the remaining languages. Consequently, relevant publications might have been excluded based on language.

The quality of reporting in included studies is lacking in many cases. Missing information for reproducing the methods of the studies was identified, and only 13 out of 39 studies included in Database 2 did not miss at least a high-level description of all indexed characteristics.

This systematic map is not intended to provide a definitive answer to the question of how to set up the perfect OB-OC co-culture. Instead, it allows searching through all relevant co-culture studies looking for specific matching experimental characteristics or culture details that may be applicable to one's own research. For this, it contains the possibility to search, sort and filter through many relevant characteristics. This allows one to find relevant studies that may have already (partly) studied one's research question, or that can be used as a guide to design comparable experiments.

## Conclusion

With this systematic map, we have generated an overview of existing OB-OC co-culture studies published until January 6, 2020, their methods, predetermined outcome measures (formation, resorption, ALP and TRAP quantification), and other useful parameters for analysis. The two constructed databases are intended to allow researchers to quickly identify publications relevant to their specific needs, which otherwise would have not been easily available or findable. The presented high-level evaluation and discussion of the major extracted methodological details provides important background information and context, suggestions and considerations covering most of the used cell sources, culture conditions and methods of analysis. Finally, this map includes the instructions for others to expand and manipulate the databases to answer their own more specific research questions.

## Supporting information

**S1 File. Database 1.** This database contains all studies in which at least one relevant outcome measure was investigated. Characteristics of outcome measures and descriptive statistics are listed in this database.
(XLSM)

**S2 File. Database 2.** This database contains all studies in which at least one relevant outcome measure was investigated for both OB and OC. Characteristics of cells, methods and culture conditions, and descriptive statistics are listed in this database.
(XLSM)

**S3 File. Using the databases.** This document provides instructions on how to operate the databases, how to add publications and expand the analyses with more elements.
(DOCX)

**S4 File. List of all OB-OC co-cultures.** This list contains the initial list of 694 OB-OC cocultures obtained after screening, before full-text investigation and exclusion based on outcome measures.
(XLSX)

**S5 File. PRISMA checklist.** The PRISMA checklist describing all elements of the systematic review, and on what page or which section of the submitted manuscript to find them.
(PDF)

**S6 File. Systematic review protocol and search queries.** The protocol and search queries as they were published prior to execution of the fulltext screening phase.
(PDF)

## Author Contributions

**Conceptualization:** Stefan J. A. Remmers, Rob B. M. de Vries, Sandra Hofmann.

**Data curation:** Stefan J. A. Remmers, Bregje W. M. de Wildt, Michelle A. M. Vis, Eva S. R. Spaander.

**Formal analysis:** Stefan J. A. Remmers, Bregje W. M. de Wildt, Michelle A. M. Vis, Eva S. R. Spaander.

**Funding acquisition:** Stefan J. A. Remmers, Rob B. M. de Vries, Keita Ito, Sandra Hofmann.

**Investigation:** Stefan J. A. Remmers, Bregje W. M. de Wildt, Michelle A. M. Vis.

**Methodology:** Stefan J. A. Remmers, Rob B. M. de Vries, Sandra Hofmann.

**Project administration:** Stefan J. A. Remmers, Sandra Hofmann.

**Resources:** Stefan J. A. Remmers, Keita Ito, Sandra Hofmann.

**Software:** Stefan J. A. Remmers.

**Supervision:** Stefan J. A. Remmers, Bregje W. M. de Wildt, Sandra Hofmann.

**Validation:** Stefan J. A. Remmers, Rob B. M. de Vries.

**Visualization:** Stefan J. A. Remmers, Bregje W. M. de Wildt.

**Writing – original draft:** Stefan J. A. Remmers.

**Writing – review & editing:** Stefan J. A. Remmers, Bregje W. M. de Wildt, Michelle A. M. Vis, Rob B. M. de Vries, Keita Ito, Sandra Hofmann.

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
