## [Decision Letter · Decision Letter 0]

23 Sep 2021

PONE-D-21-28691Osteoblast-osteoclast co-cultures: a systematic review and map of available literaturePLOS ONE

Dear Dr. Remmers,

Thank you for submitting your manuscript to PLOS ONE. After careful consideration, we feel that it has merit but does not fully meet PLOS ONE’s publication criteria as it currently stands. Therefore, we invite you to submit a revised version of the manuscript that addresses the points raised during the review process.

This manuscript has been found of interest, pending minor revision.In particular the Authors must:-reduce the length  of the manuscript because it is dispersive and several concepts and sentences are redundant;-the fluency of the overall manuscript must be revised: it is now not fluent and difficult for readers.==============================

We look forward to receiving your revised manuscript.

Kind regards,

Gianpaolo Papaccio, M.D., Ph.D.

Academic Editor

PLOS ONE

Journal Requirements:

Reviewers' comments:

Reviewer's Responses to Questions

**Comments to the Author**

1. Is the manuscript technically sound, and do the data support the conclusions?

Reviewer #1: Yes

Reviewer #2: Yes

2. Has the statistical analysis been performed appropriately and rigorously? 

Reviewer #1: N/A

Reviewer #2: N/A

3. Have the authors made all data underlying the findings in their manuscript fully available?

Reviewer #1: Yes

Reviewer #2: Yes

4. Is the manuscript presented in an intelligible fashion and written in standard English?

Reviewer #1: Yes

Reviewer #2: Yes

5. Review Comments to the Author

Reviewer #1: In this review Authors aim to create databases to facilitate research of methods concerning co-culture of OB and OC of all papers published up to 6 January 2020.

The idea is very useful and the review is well organized. I any case, Authors should reduce it because it is too long and dispersive

Reviewer #2: The review is very interesting. The Authors made a tremendous work of research in scientific literature. It is well organized and can help researchers to choose the best protocol for their specific experimental demands. Although this, the Authors must make the manuscript more fluent.

It is too long and ,in such points, redundant.

6. PLOS authors have the option to publish the peer review history of their article (what does this mean?). If published, this will include your full peer review and any attached files.

Reviewer #1: No

Reviewer #2: No

---

## [Author Response · Author response to Decision Letter 0]

20 Oct 2021

Response to Reviewers:

The authors thank the reviewers for their positive approach to this manuscript and agree with their concerns regarding length and fluency. The manuscript has now been thoroughly shortened and reduced in length by more than 25%. This was achieved mainly by removing redundant or duplicate information and carefully rewriting each section of the manuscript in a concise manner while still describing the important content. We believe we have succeeded in improving readability and fluency, without significantly reducing the value that the manuscript provides.

---

## [Editor Report · Decision Letter 1]

22 Oct 2021

Osteoblast-osteoclast co-cultures: a systematic review and map of available literature

PONE-D-21-28691R1

Dear Dr. Remmers,

We’re pleased to inform you that your manuscript has been judged scientifically suitable for publication and will be formally accepted for publication once it meets all outstanding technical requirements.

Kind regards,

Gianpaolo Papaccio, M.D., Ph.D.

Academic Editor

PLOS ONE

Additional Editor Comments (optional):

The Authors have addressed the previous criticisms
---

## [Editor Report · Acceptance letter]

27 Oct 2021

PONE-D-21-28691R1 

Osteoblast-osteoclast co-cultures: a systematic review and map of available literature 

Dear Dr. Remmers:

I'm pleased to inform you that your manuscript has been deemed suitable for publication in PLOS ONE. Congratulations! Your manuscript is now with our production department. 

Kind regards, 

on behalf of

Prof. Gianpaolo Papaccio 

Academic Editor

PLOS ONE